# A Modified ^1^H-NMR Quantification Method of Ephedrine Alkaloids in Ephedrae Herba Samples

**DOI:** 10.3390/ijms241411272

**Published:** 2023-07-10

**Authors:** Yue-Chiun Li, Chia-Hung Wu, Thi Ha Le, Qingjun Yuan, Luqi Huang, Guo-Fen Chen, Mei-Lin Yang, Sio-Hong Lam, Hsin-Yi Hung, Handong Sun, Yi-Hung Wu, Ping-Chung Kuo, Tian-Shung Wu

**Affiliations:** 1School of Pharmacy, College of Medicine, National Cheng Kung University, Tainan 701, Taiwan; ycli0126@gmail.com (Y.-C.L.); lethiha.hup@gmail.com (T.H.L.); l3891104@nckualumni.org.tw (M.-L.Y.); shlam@mail.ncku.edu.tw (S.-H.L.); z10308005@email.ncku.edu.tw (H.-Y.H.); 2School of Post-Baccalaureate Chinese Medicine, China Medical University, Taichung 404, Taiwan; 108030049@365.cmu.edu.tw; 3State Key Laboratory Breeding Base of Dao-di Herbs, National Resource Center for Chinese Materia Medica, China Academy of Chinese Medical Sciences, Beijing 100010, China; yuanqingjun@icmm.ac.cn (Q.Y.); huangluqi01@126.com (L.H.); 4Department of Chemistry, National Chung-Hsing University, Taichung 402, Taiwan; edward.chen@jeolanalytical.com.tw; 5Kunming Institute of Botany, Chinese Academy of Sciences, Kunming 650201, China; hdsun@mail.kib.ac.cn; 6Hsinhua Forest Station, The Experimental Forest Management Office, National Chung-Hsing University, Taichung 402, Taiwan; yihung@dragon.nchu.edu.tw

**Keywords:** ephedrine alkaloid, Ephedrae Herba, cyclized derivative, quantitative nuclear magnetic resonance spectroscopy (qNMR), two-dimensional NMR (2D NMR)

## Abstract

A previous ^1^H-NMR method allowed the quantification of ephedrine alkaloids; however, there were some disadvantages. The cyclized derivatives resulted from the impurities of diethyl ether were identified and benzene was selected as the better extraction solvent. The locations of ephedrine alkaloids were confirmed with 2D NMR. Therefore, a specific ^1^H-NMR method has been modified for the quantification of ephedrine alkaloids. Accordingly, twenty Ephedrae Herba samples could be classified into three classes: (I) *E. sinica*-like species; (II) *E. intermedia*-like species; (III) others (lower alkaloid contents). The results indicated that ephedrine and pseudoephedrine are the major alkaloids in *Ephedra* plants, but the concentrations vary greatly determined by the plant species and the collection locations.

## 1. Introduction

The genus of *Ephedra* belongs to the Ephedraceae family and contains more than 60 species which are widely distributed in the arid and semi-arid regions of Asia, Europe, Northern Africa (Sahara), Southwestern North America, and South America [1]. On the other hand, it is reported that there are 12 *Ephedra* species growing in China [2,3]. According to the Chinese Pharmacopeia, Ephedrae Herba is derived mainly from the aerial parts of *E. sinica*, *E. equisetina*, and *E. intermedia* [4,5,6]. Ephedrae Herba has been used in traditional Chinese medicine (TCM) for a long time as a diaphoretic, stimulant, and antiasthmatic. In addition, it could be used to treat bronchitis, acute nephritic edema, cough, and asthma, to induce perspiration, and to reduce fever [5,6].

The medicinal properties of *Ephedra* species could be classified based on the contents of ephedrine alkaloids, such as methylephedrine (ME) (**1**), ephedrine (EP) (**2**), norephedrine (NE) (**3**), norpseudoephedrine (NP) (**4**), methylpseudoephedrine (MP) (**5**), and pseudoephedrine (PE) (**6**) (structures as shown in Figure 1) [5,6]. It was reported that total ephedrine alkaloids found in the aerial parts of *Ephedra* species range from 0.02 to 3.4% [5]. The highest alkaloid content is found in *E. equisetina* (2.7 ± 0.6%), followed by *E. intermedia* (1.5 ± 0.7%) and *E. sinica* (1.4 ± 0.6%) [4]. *E. sinica* and *E. equisetina* mainly consist of *l*-ephedrine, whereas *E. intermedia* mainly contains *d*-pseudoephedrine [4]. In addition, the diverse geographical origins of the plant materials result in the total content of the main active alkaloids differing substantially from species to species [2,4].

The pharmacological and toxicological effects of Ephedrae Herba depend on the individual ephedrine alkaloid type, its enantiomeric form, and receptor binding characteristics. Ephedrine (EP) is a sympathomimetic agonist at both the *α-* and *β*-adrenergic receptors, leading to an increased cardiac rate and contractility, peripheral vasoconstriction, bronchodilation, and central nervous system (CNS) stimulation [1,5]. The effects of vasoconstriction and bronchodilation explain the traditional use of *Ephedra* as a nasal decongestant and an anti-asthmatic [5]. However, EP enhances the release of catecholamines, thus triggering side effects on the cardiovascular system [1,7]. In recent years, dietary supplements containing Ephedrae Herba have been sold extensively for the treatment of obesity or for increasing performance in body building. These usages may be due to the CNS stimulation and thermogenic properties of EP [5]. Although EP does suppress appetite, the main mechanism for promoting weight loss appears to be by increasing the metabolic rate of adipose tissue [5]. Pseudoephedrine (PE) acts similarly, but with fewer CNS effects. Therefore, PE may be relatively safe for the treatment of non-chronic nasal congestion [1,8]. According to the Food and Drug Administration (FDA) assessment conducted in December 2003, dietary supplements that contain ephedrine alkaloids represent an unacceptable health risk. Consequently, the FDA banned usage of ephedrine alkaloids (regardless of their botanical origin) in dietary supplements [5,7]. The International Olympic Committee listed ephedrine and related compounds as stimulants in athletic sports in 2003. However, it is known that Ephedrae Herba products are often used for the treatment of colds; the International Athletic Committee has adopted a quantitative limitation for ephedrine alkaloids in urine. An athlete is regarded as “positive” for pseudoephedrine and norephedrine at 25 μg/mL, for ephedrine at 10 μg/mL, and for norpseudoephedrine at 5 μg/mL [9].

The regulated botanical origins of Chinese crude drug Mahuang (Ephedrae Herba) have been rapidly depleted because of habitat destruction and overharvesting [10]. Other *Ephedra* species that are not in the official pharmacopoeia, such as *E. gerardiana*, *E. likiangensis*, *E. przewalskii*, and *E. minuta*, are also used as Mahuang [11]. However, their reputation is not as good as that of the *Ephedra* species listed in the pharmacopoeia of China, and their ephedrine contents are usually lower according to the reports [11]. Therefore, determining the contents of ephedrine alkaloids in different *Ephedra* species is important to assess the quality of the crude drug and safe medication. Numerous analytical methods, such as thin-layer chromatography (TLC) [12], capillary electrophoresis (CE) [13], high-performance liquid chromatography (HPLC) [14,15,16,17,18,19,20,21,22,23], gas chromatography (GC) [9], and GC-mass spectrometry (GC-MS) [24,25], have been applied to the quantitative analysis of *Ephedra* alkaloids. Regarding the HPLC methods, the reversed-phase separation of basic analytes such as ephedrine alkaloids often results in broad and tailing bands that are caused by acidic sites on the column packing. The GC method has successfully separated and identified all six alkaloids. However, this method requires complex cleanup procedures and precolumn derivatization before analysis, leading to time-consuming protocols [26]. Hence, a simple, sensitive, accurate, and rapid method for the simultaneous identification and determination of *Ephedra* alkaloids for the quality control of *Ephedra* raw materials and commercial pharmaceutical prescriptions is urgently required.

High-resolution nuclear magnetic resonance (NMR) spectroscopy is regarded as a potent tool for the quality control of phytochemical preparations [27,28,29,30,31], clinical diagnosis [32], and monitoring of treatment [32]. Compared to classic analytical methods, the quantitative NMR (qNMR) offers several advantages. It allows for a rapid method implementation and simultaneous quantification of several metabolites without the standard compounds and any sample pretreatment steps. It is noninvasive, rapid, and does not require any sample pre-clean steps. In addition, it is not necessary to prepare the calibration curves based on standard compounds, and it monitors the constituents presented in herbal preparations simultaneously in a single analysis. In 2003, Kim et al. used qNMR for the quantitative analysis of four ephedrine analogues from *Ephedra* species including EP, PE, ME, and MP without any pre-cleaning steps [33]. Using this method, the contents of the ephedrine alkaloids can be analyzed within much shorter timeframes than the conventional chromatographic methods; however, in Kim et al., the determination was limited to four of the six *Ephedra* alkaloids. In 2021, Hung et. al. described a ^1^H-NMR spectroscopic method for the quantitative analysis of six ephedrine alkaloid derivatives in *Ephedra* species and related commercial traditional Chinese medicine prescriptions [34]. However, there were still some disadvantages associated with this method. First, some unusual derivatives were observed due to the utilization of ether. Second, the locations of some ephedrine alkaloid signals were confusing with ^1^H-NMR only. Therefore, in the present study, this method is further modified and utilized for the quantitative analysis of Ephedrae Herba samples collected from different regions. Hopefully, the developed method could be applied as the standard for the quantification of ephedrine alkaloids in the near future.

## 2. Results

### 2.1. Identification of the Cyclization Products

In our previous report, diethyl ether was selected for extraction of the ephedrine alkaloids; however, it would result in the cyclization products (Figure 2). In addition to the common ephedrine (**2**) and norephedrine (**3**), there were three characterized compounds with proton signals observed at *δ*_H_ 5.14 (**7**), 4.99 (**8**), 4.53 (**9**) ppm in the *E. sinica* extract (EP01), and these cyclization products could also be detected in the extracts of *E. intermedia* (EP03) and *E. equisetina* (EP02) (Appendix A). These products may be the result of the chemical reactions among the ephedrine alkaloids and the impurities of diethyl ether. The chemical structures of these cyclized ephedrine alkaloids were further determined by spectrometric and spectroscopic analyses.

The molecular formula of compound **7** was deduced to be C_17_H_19_NO according to its ESI-TOF-MS analytical data (*m*/*z* 254.1548 for [M+H]^+^, calcd. for C_17_H_20_NO, 254.1545, error = 1.23 ppm) (Appendix A). Through the ^1^H-NMR, ^13^C-NMR, HMBC (Appendix A), and HSQC-TOCSY (Figure 3) analyses, the structure of **7** was established to be (2*R*,4*R*,5*S*)-3,4-dimethyl-2,5-diphenyloxazolidine by comparison of its spectral characteristics with the literature data [35]. In addition, ephedrine (**2**) [36] and norephedrine (**3**) [37] could be confirmed with similar protocols. The molecular formula of compounds **8** and **9** were determined as C_12_H_17_NO, deduced from its ESI-TOF-MS analytical data (*m*/*z* 192.1380 for [M+H]^+^, calcd. for C_12_H_18_NO, 192.1388, error = −4.55 ppm) (Appendix A). Their structures were elucidated according to the spectroscopic analytical data (Appendix A), and the HSQC-TOCSY spectra are shown in Figure 4 and Figure 5. In the total correlation spectroscopy (TOCSY) experiment, correlations between all protons within a given spin system were detected. Correlations are observed between remote protons as long as there are couplings between every intervening hydrogens [38]. It is very useful to identify protons of amino acids and sugars, including large molecules, such as peptides, proteins, and polysaccharides [38]. However, one of the disadvantages of TOCSY is the overlap between peaks close to one another. Heteronuclear single quantum coherence (HSQC)-TOCSY solves the major problem through resolution of crosspeaks into the ^13^C dimension and allows for easier characterization [39,40]. In the present case, it is very difficult to assign a chemical structure to each cyclized ephedrine alkaloid in crude extracts. After HSQC-TOCSY analysis, the fragments of rings from cyclized ephedrine alkaloids could be provided (Figure 4 and Figure 5), and the planar structures were supported by the heteronuclear multiple bond correlation (HMBC) spectrum (Appendix A). These structures could be further verified through their characteristic carbon signals (Appendix A). Therefore, **8** and **9** were identified as (2*R*,4*R*,5*S*)-2,3,4-trimethyl-5-phenyloxazolidine [35] and (2*R*,4*R*,5*R*)-2,3,4-trimethyl-5-phenyloxazolidine [35] (Figure 2), respectively, by comparison of their spectral data with those reported in the literature. The complete list of proton and carbon signals of **7**–**9** are provided in Table 1.

Chloroform was also used for extraction, but it led to more impurities in the extracts (Appendix A). It produced more complex signals in the observed region for H-1 of ephedrine alkaloids at *δ* 4.0–5.0 ppm. Therefore, in the present study, benzene is selected as the extraction medium since few interferences at *δ* 4.0–5.0 ppm were observed and the resulting extracts were more pure (Appendix A).

### 2.2. Quantification of the Ephedrine Alkaloids

In our previous report [34] for the quantitative analysis of the six *Ephedra* alkaloids, i.e., methylephedrine (ME) (**1**), ephedrine (EP) (**2**), norephedrine (NE) (**3**), norpseudoephedrine (NP) (**4**), pseudoephedrine (PE) (**5**), and methylpseudoephedrine (MP) (**6**), H-1 was selected as a target signal in the ^1^H-NMR spectra because these signals of the ephedrine alkaloids have separate resonances and do not overlap with other signals from the extracts. Kim et al. recorded the ^1^H-NMR spectrum of four standards (ME, EP, MP, and PE) and analyzed their contents in *Ephedra* samples according to the chemical shifts in the standard spectrum [33]. However, some peaks would shift a little and even exchange their locations with neighbor alkaloids in different samples due to the interference from other constituents in the mixture. Thus, it is not easy to identify or quantify these alkaloids in these samples with ^1^H-NMR only. To ensure a correct NMR signal identification for each alkaloid, ^1^H-, HSQC-TOCSY and HMBC experiments were conducted to determine their positions (Appendix A). Traditionally, TOCSY is very useful to identify protons of polymers, but often the overlapped peaks render distinction challenging [33]. To solve the problem, 2D-TOCSY is often utilized, for example, the ^13^C dimension of HSQC render distinction easier and more accurate (Appendix A). After the HSQC-TOCSY analysis of EP01, the H-1 of each alkaloid was determined and the quantification of corresponding ephedrine alkaloids was conducted.

Considering of the solubility of ephedrine alkaloids, CDCl_3_ was used as the solvent to ensure that all the extracts could be dissolved. The H-1 proton signal of each ephedrine alkaloid observed in the range of *δ* 4.0–5.0 ppm (doublet) was quite well separated from the others and did not overlap with other signals from the extract in the ^1^H-NMR spectrum (Figure 6). According to the literature reports and our 2D NMR analysis, the target protons (H-1) of EP and PE in the present study were resonating at *δ* 4.71 (d, *J* = 3.6 Hz) and 4.12 (d, *J* = 8.4 Hz), respectively. The H-1 proton signals of ME and NP are also well separated from each other and observed at *δ* 4.90 (d, *J* = 4.0 Hz) and 4.18 (d, *J* = 6.8 Hz), respectively. For two further alkaloids, namely NE and MP, the H-1 signals should be resonating close to *δ* 4.48 and 4.14; however, these peaks were not observed in this sample. Therefore, these data suggest that the H-1 signal is suitable for use as a target peak for quantification. Our results also highlight mistakes in the previous publication [33]. Although the identification of six alkaloids in Ephedrae Herba samples were unambiguous with the assisatance of 2D NMR experiments, routine determinations in other samples were achieved simply by comparison of chemical shift and coupling constants of H-1 signals of these alkaloids.

Twenty samples of Ephedrae Herba plants were collected from different locations in Taiwan and China. In total, eleven species of Ephedrae Herba were sampled among the commonly used medicinal plant materials in traditional Chinese medicines (Table 2). Powdered Ephedrae Herba was extracted according to the protocols described in the experimental section; the resulting weights of samples are listed in Appendix A. A suitable internal standard should preferably be a stable compound with a signal in a non-crowded region of the ^1^H-NMR spectrum. Therefore, anthracene with a signal at *δ* 8.26 and the integral values remaining constant throughout 48 h was selected and added into the herbal extract samples. The ^1^H NMR spectra of these samples are shown in Appendix A. In the case of the qNMR analysis, the amount of each alkaloid was calculated by the relative ratio of the intensity of H-1 signal to the known amount of anthracene with the Equation (1) (See Section 4.5) and the results of the ME, EP, NE, NP, and PE contents in these samples are shown in Table 3. The average recovery of EP was about 95% (in triplicate), and the LOD for EP under the present experimental parameter was 0.05 mg/mL.

## 3. Discussion

Various impurities including formaldehyde, acetaldehyde, and propionaldehyde were observed in diethyl ether even with the analytical grade chemicals [41]. It was reported that ephedrine alkaloids would be degraded under reaction with aliphatic and aromatic aldehydes [41,42,43,44,45,46]. Therefore, ephedrine derivatives were transformed into oxazolidines after extraction by diethyl ether. These cyclized products were determined by MS and NMR analysis (Figure 3, Figure 4, Figure 5 and Appendix A). In the present research, benzene is selected as the extraction solvent since the extracted alkaloids were more pure than those extracted by diethyl ether (Appendix A). Although the contents of total ephedrine alkaloids are lower compared with the previous report [34], our recovery tests showed promising results for the present method. For example, in *E. sinica*, the content of ephedrine was reported as 8.49 mg/g, but in this study it was only 4.33 mg/g. Other *Ephedra* alkaloids also display the same tendency (Table 3). In the present qNMR results, total alkaloids contents were lower than those determined by HPLC [22]. It may be due to differences inherent to the plant species and collection locations. Nevertheless, the relative contents of *Ephedra* alkaloids could still be determined for comparison among different *Ephedra* species; the lower yields are not so relevant. Extraction with diethyl ether would definitely result in the cyllization products despite the higher extraction efficiency. Therefore, benzene was used for further extraction of other *Ephedra* plants and the obtained extracts were quantified by NMR analysis. Furthermore, no ephedrine alkaloids were observed in the residual water layers, since almost all alkaloid molecules were extracted from the organic layer under the present extraction conditions.

In total, twenty Ephedrae Herba samples were collected from different locations in Taiwan and China. After extraction following the present protocols, the obtained extracts were analyzed by the ^1^H-NMR spectroscopic method. Quantification results are presented in Table 3. The results indicate that ephedrine (EP) (**2**) and pseudoephedrine (PE) (**6**) are the main alkaloids in these *Ephedra* plants, but the concentrations of the six ephedrine alkaloids vary greatly with plant species and collection locations. Among the examined samples, eight showed total alkaloids content higher than 3 mg/g,; all of these were found to belong to *E. sinica*, *E. equisetina*, *E. intermedia*, *E. monosperma*, and *E. saxatilis*. The alkaloid contents of *E. glauca* and *E. gerardiana* are somewhat lower, ranging from 1 to 3 mg/g of dried plant material weight. Moreover, some samples did not contain these alkaloids, such as *E. lepidosperma*, *E. minuta*, *E. przewalskii*, and *E. regeliana*. 

EP (**2**) was the main component in the tested samples derived from *E. sinica, E. equisetina, E. gerardiana*, and *E. saxatilis*. In contrast, in *E. intermedia, E. glauca, E. przewalskii*, and *E. monosperma*, the content of PE (**6**) was higher than that of EP (**2**). According to the Chinese pharmacopeia, Ephedrae Herba (Mahuang) is derived mainly from the aerial parts of *E. sinica*, *E. equisetina*, and *E. intermedia* [5]. Unfortunately, in China, the resources on which these three species depend have been rapidly depleted due to habitat destruction and overharvesting [10]. Based on the present analytical results, EP- and PE-rich species such as *E. monosperma*, *E. saxatilis*, *E. glauca,* and *E. gerardiana* could also be used as alternatives to Mahuang, even though they are not compiled in the official pharmacopoeia. According to our experimental quantification results, Ephedrae Herba samples could be classified into three classes: (I) *E. sinica*-like species (EP > PE), including samples EP01, EP02, EP04, EP11, EP17; (II) *E. intermedia*-like species (PE > EP), including samples EP03, EP08, EP10, EP12, EP16, EP18; and (III) others (lower alkaloid contents). Among the samples of Class I, EP01 contained the highest EP levels (4.33 mg/g), and EP11 presented the lowest (1.55 mg/g). Among the samples of Class II, PE was the richest in EP16 (4.41 mg/g); on the contrary, EP12 showed the lowest levels of PE (1.99 mg/g). Moreover, samples EP05, EP06, EP07, EP09, EP13, EP14, EP15, EP19, and EP20 (Class III) displayed very low alkaloid contents, with some alkaloids not detected by the present method. Kajimura et al. reported that alkaloid content was low in the early stages of stem growth [47]. Therefore, the present results may be attributed to the complex environment factors, mainly the growth periods. Furthermore, the previous report also indicates that *E. przewalskii* and *E. lepidosperma* contained few alkaloids [2]. The present results indicate that EP06 (*E. intermedia*), EP07, EP09 (*E. przewalskii*), EP13 (*E. lepidosperma*), EP14 (*E. minuta*), EP15 (*E. regeliana*), EP19, and EP20 did not show a significant presence of ephedrine alkaloids. Therefore, they were not suitable for use as an alternative to Mahuang.

Among the Class II samples, EP10 and EP18 were peculiar, since they were authenticated as *E. equisetina* and *E. sinica*, respectively. Based on the chemical and analytical results, they were classifed as *E. intermedia*-like species, since they exhibited higher levels of PE than EP. These results indicate that the two Ephedrae Herba samples may have been identified incorrectly. Based on these quantification data, the present analytical method could be considered as a feasible tool for quality control of commercial Mahuang products.

## 4. Materials and Methods

### 4.1. General

All the chemicals, unless specifically indicated otherwise, were purchased from Merck KGaA (Darmstadt, Germany). Anthracene (97.0%) and CDCl_3_ (99.9%) were obtained from Sigma-Aldrich (St. Louis, MO, USA). With respect to analyses, 2D NMR were recorded on the Jeol ECZ600R/S1 600 MHz (Jeol, Tokyo, Japan) and Agilent DD2 600 MHz (Agilent, Santa Clara, CA, USA), and quantitative ^1^H-NMR analyses were performed on the Bruker AV-400 400 MHz (Bruker, Billerica, MA, USA) NMR spectrometers with tetramethylsilane as the internal standard. Chemical shifts are reported in parts per million (ppm, *δ*). High resolution electrospray ionization mass spectrometry (HR-ESI-MS) were examined on a JEOL JMS-700 spectrometer (Jeol, Tokyo, Japan) that the experimental data were afforded in the positive-ion mode.

### 4.2. Plant Material and Authentication

Ephedrae Herba plants were collected in various regions and identified as shown in Table 2. The voucher specimen was deposited in the herbarium of the School of Pharmacy, National Cheng Kung University, Tainan, Taiwan.

### 4.3. 2D Analysis of the Ephedrine Alkaloids and Their Cyclization Derivatives 

The powdered Ephedrae Herba samples were extracted following our previously reported protocols [34]. The resulted cyclization derivatives were observed in their ^1^H-NMR spectra (Appendix A). The molecular formula of cyclization derivatives were determined by HR-ESI-MS analytical data. ^1^H-, ^13^C-, HSQC-TOCSY, HMBC NMR analyses of the ephedrine alkaloids and cyclization products were recorded on the Jeol ECZ600R/S1 600 MHz NMR spectrometer with the Royal 5 mm probe.

### 4.4. Sample Extraction for qNMR Analysis

The extraction of alkaloids was performed according to the reported method [34] with minor modifications. Powdered Ephedrae Herba samples (~2 g) were immersed in 50 mL 0.5% HCl aqueous solution and extracted by ultrasonicator at 60 °C. After 30 min, the infusions were filtered through a suction filter. Each sample was extracted four times, and the pH of the combined solution was adjusted to 10.0 by Na_2_CO_3_; the suspension was successively obtained after centrifugation at 3000 rpm for 10 min. NaCl was added to the suspension until saturation and the suspension was then centrifuged at 3000 rpm for a further 10 min. The yielded upper layer was partitioned with benzene five times, and the combined organic solution was concentrated in vacuum to produce the Ephedrae Herba extract (Appendix A).

### 4.5. qNMR Analysis of Ephedrine Alkaloids

Ephedrae Herba extracts and internal standard (anthracene) were dissolved in 0.6 mL CDCl_3_ and analyzed by Bruker AV-400 NMR spectrometer. One hundred scans were recorded in FID resolution, 0.39 Hz/point; spectrum digital resolution, 0.26 Hz/point; zero filling, 16,384; no digital filtering (apodization); spectra width, 6394 Hz; a 90 pulse was used to obtain the maximum sensitivity; relaxation delay, 20 s; acquisition time, 2.56 s. For quantitative analysis, the peak area of H-1 signals of these compounds (Table 4) were utilized since they were well separated in the region of δ 4.0–5.0 ppm, and the start and end points of the integration of each peak were selected manually. The alkaloid contents were calculated as followed.
(1)$$\mathrm{Alkaloid}\;\mathrm{contents}\;(\mathrm{mg}/g)=\frac{2\times \;A\;\times \;C\;\times \;M}{B\;\times \;D}$$

A: Peak area of H-1 signal for each alkaloid (1H)

B: Peak area of anthracene (*δ*_H_ 8.36, 2H)

C: The concentration of anthracene (M)

D: The weight of herba extract

M: Molecular weight of each alkaloids

**Table 4 ijms-24-11272-t004:** ^1^H NMR signal of H-1 for the ephedrine alkaloids (δ in ppm).

Compound	H-1
methylephedrine (ME) (**1**)	4.96
ephedrine (EP) (**2**)	4.76
norephedrine (NE) (**3**)	4.52
norpseudoephedrine (NP) (**4**)	4.24
methylpseudoephedrine (MP) (**5**)	4.19
pseudoephedrine (PE) (**6**)	4.17

### 4.6. Recovery, Limit of Quantification (LOQ), and Limit of Detection (LOD) of Ephedrine

Recovery tests were selected to determine the accuracy of the method, in which three different concentrations of ephedrine (0.5, 1.0, 2.0 mg, respectively) were added to the sample and the recovery percentages were calculated using the measured contents divided by the contents of added standards and original sample obtained by ^1^H NMR analysis. A blank recovery sample was prepared and analyzed for the comparison. LOQ and LOD for ephedrine under the present NMR analytical conditions (100 scans) were determined at the signal-to-noise ratios of 10 and 3, respectively.

### 4.7. Characterization of Ephedrine Alkaloids by 2D NMR Analysis

The Ephedrae Herba extract was redissolved in CDCl_3_ and subjected to Agilent DD2 600 MHz for 2D analysis. For HMBC analysis, 64 scans were performed; the number of increments was 200. For HSQC-TOCSY examination, each experiment was scanned for 84 times; the number of increments was 128; mixing time was 80 ms. JEOL Delta V5 software was used for data processing.

## 5. Conclusions

In the present study, a modified ^1^H-NMR method is implemented successfully for the determination of six ephedrine alkaloids (**1**–**6**) in 20 samples of Ephedrae Herba collected from different regions across Taiwan and China. Benzene is selected as the better solvent for extraction of ephedrine alkaloids due to fewer interferences and no cyclization products. Furthermore, the locations of ephedrine alkaloid signals are unambiguously determined by 2D NMR analysis. The present method retains all the advantages of qNMR, including the rapid and simultaneous quantification of several targets, without the standard compounds and any sample pretreatment steps.

According to our experimental quantification results, the examined Ephedrae Herba samples could be divided into three classes: (I) *E. sinica*-like species (EP > PE), (II) *E. intermedia*-like species (PE > EP) and (III) others (lower alkaloid contents). The results showed that the main alkaloids in these *Ephedra* plants are ephedrine (EP) (**2**) and pseudoephedrine (PE) (**6**). Are the species containing no/low *Ephedra* alkaloids (Class III) botanically related? More experiments will need to be conducted to address it in the future. The concentrations of these six ephedrine alkaloids vary greatly depending on the plant species and collection location. These experimental data could provide the information related to the applicability of various *Ephedra* plants, in which the materials with low alkaloid contents are not suitable alternatives to Ephedrae Herba.

## Figures and Tables

**Figure 1 ijms-24-11272-f001:**
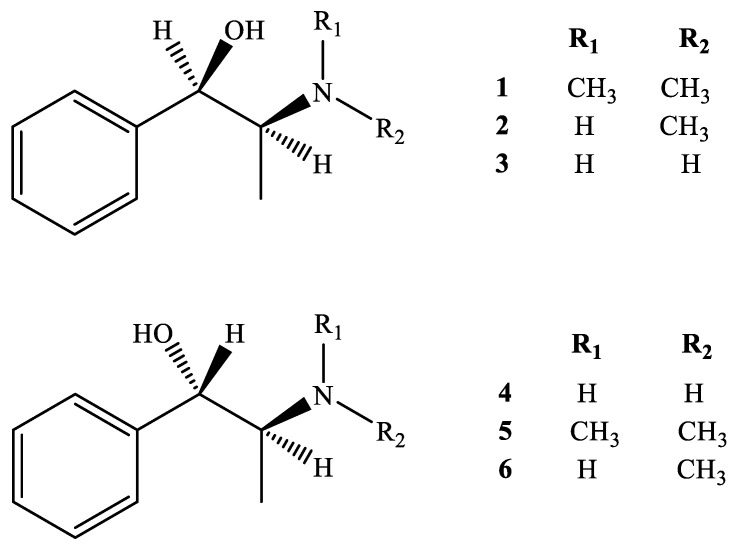
Chemical structures of **1**–**6**.

**Figure 2 ijms-24-11272-f002:**
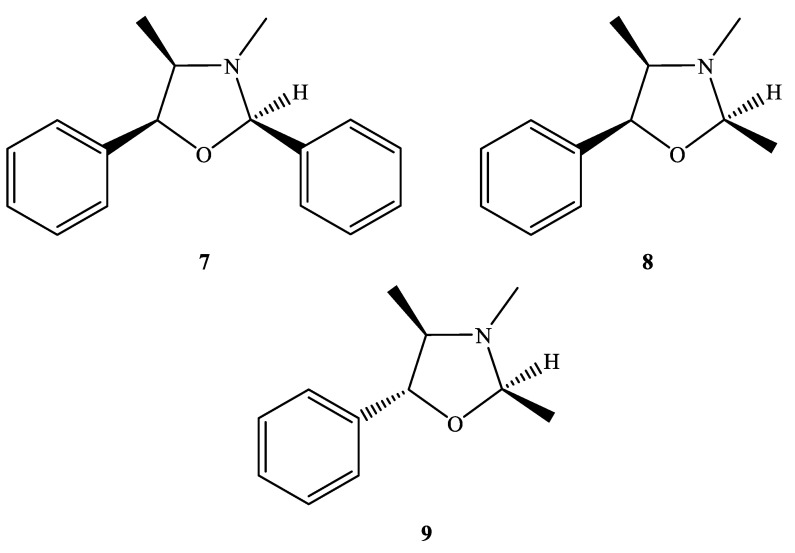
Chemical structures of **7**–**9**.

**Figure 3 ijms-24-11272-f003:**
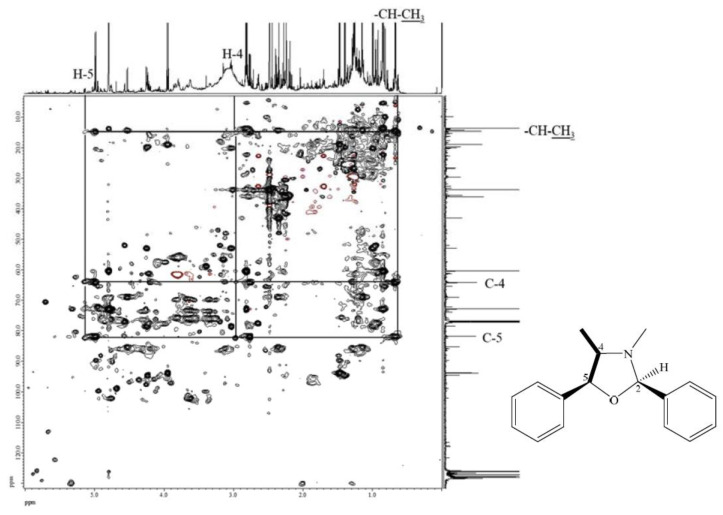
HSQC-TOCSY of compound **7**.

**Figure 4 ijms-24-11272-f004:**
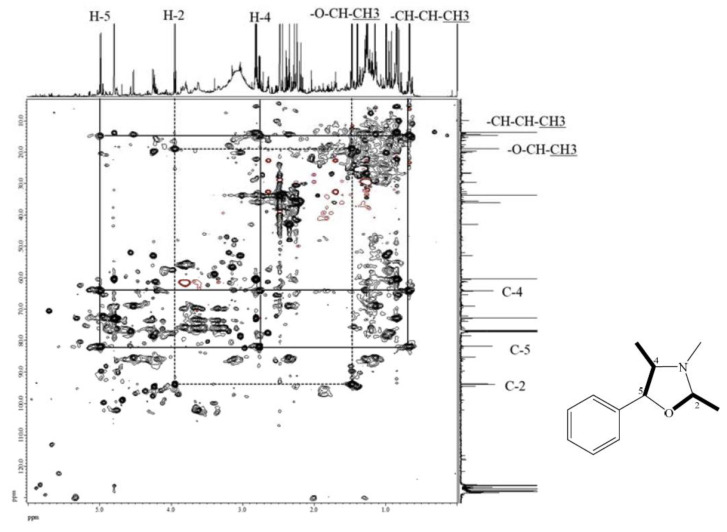
HSQC-TOCSY of compound **8**.

**Figure 5 ijms-24-11272-f005:**
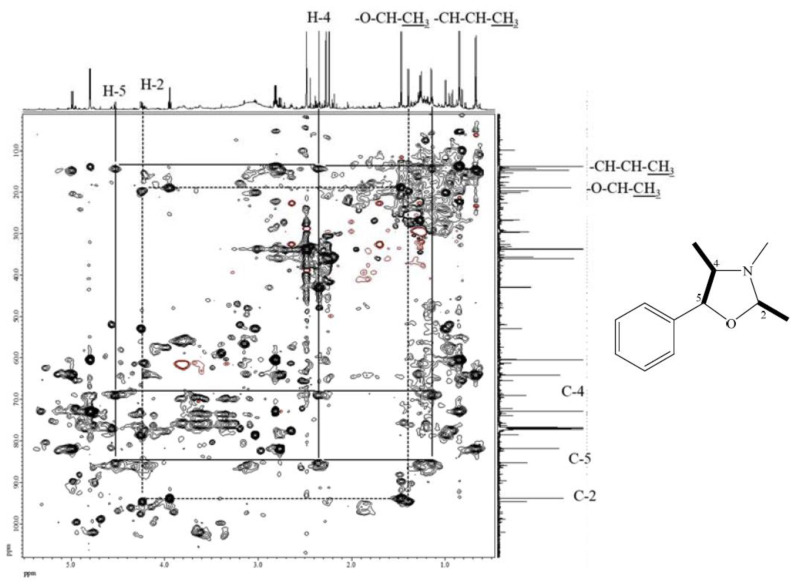
HSQC-TOCSY of compound **9**.

**Figure 6 ijms-24-11272-f006:**
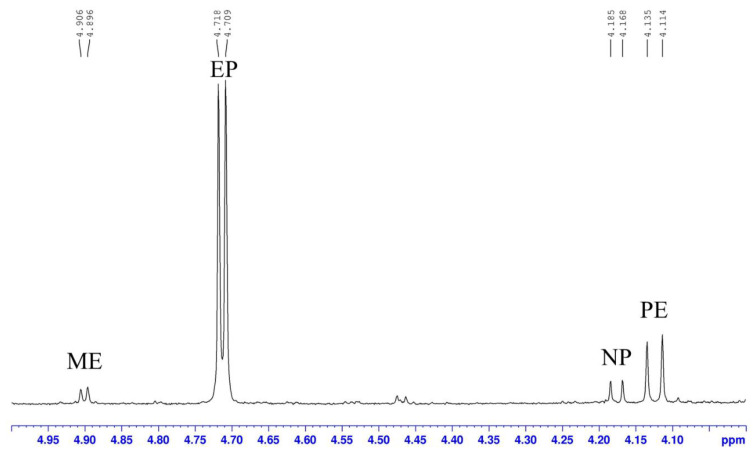
The H-1 signals of ephedrine alkaloids in the expanded ^1^H-NMR of EP01 (recorded in CDCl_3_).

**Table 1 ijms-24-11272-t001:** The key ^1^H-, and ^13^C-NMR signals of compounds **7**–**9**.

Position	7		8		9	
	*δ* _H_	*δ* _C_	*δ* _H_	*δ* _C_	*δ* _H_	*δ* _C_
2	4.69 (s, 1H)	98.8	3.95 (q, 1H, *J* = 5.4 Hz)	93.9	4.24 (dq, 1H, *J* = 5.4, 0.6 Hz)	94.6
4	2.97(m, 1H)	64.0	2.77 (dq, 1H, *J* = 8.4, 6.6 Hz)	64.2	2.36 (m)	69.1
5	5.14 (d, 1H, *J* = 8.4 Hz)	82.4	4.99 (d, 1H, *J* = 8.4 Hz)	81.9	4.53 (d, 1H, *J* = 8.4 Hz)	85.3
CH_3_-2			1.47 (d, 3H, *J* = 5.4 Hz)	18.9	1.39 (d, 3H, *J* = 5.4 Hz)	19.6
CH_3_-4	0.79 (d, 3H, *J* = 6.0 Hz)	15.0	0.67 (d, 3H, *J* = 6.6 Hz)	14.7	1.15 (d, 3H, *J* = 6.0 Hz)	14.3
N-CH_3_	2.18 (s, 3H)	35.7	2.24 (s, 3H)	36.0	2.28 (s, 3H)	35.7

**Table 2 ijms-24-11272-t002:** Twenty Ephedrae Herba samples.

Sample	Source	Collected at	Date	Authenticated by
EP01	*E. sinica*	Taiwan	February 2020	a
EP02	*E. equisetina*	Taiwan	February 2020	a
EP03	*E. intermedia*	Taiwan	February 2020	a
EP04	*E. sinica*	Shanxi	June 2020	b
EP05	*E. sinica*	Shinjang	June 2020	b
EP06	*E. intermedia*	Shinjang	June 2020	b
EP07	*E. przewalskii*	Shinjang	June 2020	b
EP08	*E. glauca*	Shinjang	June 2020	b
EP09	*E. przewalskii*	Gansu	April 2019	c
EP10	*E. equisetina*	Gansu	April 2019	c
EP11	*E. gerardiana*	Tibet	April 2019	c
EP12	*E. intermedia*	Gansu	April 2019	c
EP13	*E. lepidosperma*	Ningxia	April 2019	c
EP14	*E. minuta*	Ningxia	April 2019	c
EP15	*E. regeliana*	Shinjang	April 2019	c
EP16	*E. monosperma*	Ningxia	April 2019	c
EP17	*E. saxatilis*	Tibet	April 2019	c
EP18	*E. sinica*	Gansu	April 2019	c
EP19	*E. przewalskii*	Gansu	April 2016	c
EP20	*E. intermedia*	Gansu	April 2016	c

a, Chuang Song Zong Pharmaceutical Co., Ltd.; b, Prof. Handong Sun, Kunming Institute if Botany; c, Prof. Qingjun Yuan & Prof. Luqi Huang, National Resource Center for Chinese Materia Medica, China Academy of Chinese Medical Sciences.

**Table 3 ijms-24-11272-t003:** The contents of ephedrine alkaloids of different *Ephedra* plants.

Sample	Source	Content (mg/g Material) (RSD %)						Total Alkaloid Content (mg/g Material)	Class
		ME	EP	NE	NP	MP	PE		
EP01	*E. sinica*	0.24 (1.80)	4.33 (1.01)	-	0.32 (5.90)	-	0.89 (1.74)	5.78	I
EP02	*E. equisetina*	0.12 (2.06)	2.40 (1.14)	0.09 (1.93)	0.20 (1.72)	-	0.96 (0.65)	3.77	I
EP04	*E. sinica*	0.16 (4.10)	3.18 (0.22)	-	0.24 (4.89)	-	0.64 (2.09)	4.22	I
EP11	*E. gerardiana*	0.13 (2.03)	1.55 (0.25)	0.04 (8.67)	-	-	0.17 (4.19)	1.89	I
EP17	*E. saxatilis*	0.25 (1.14)	3.57 (0.20)	0.09 (1.30)	-	-	0.27 (3.31)	4.18	I
EP03	*E. intermedia*	-	0.30 (4.07)	-	0.50 (5.94)	-	2.74 (0.86)	3.54	II
EP08	*E. glauca*	-	0.15 (2.64)	-	0.34 (1.02)	-	2.49 (0.73)	2.98	II
EP10	*E. equisetina*	0.05 (1.95)	1.02 (0.41)	0.04 (2.52)	0.46 (1.83)	-	2.03 (0.67)	3.60	II
EP12	*E. intermedia*	-	0.24 (1.24)	-	0.28 (1.19)	-	1.99 (1.98)	2.51	II
EP16	*E. monosperma*	-	-	-	-	-	4.41 (0.19)	4.41	II
EP18	*E. sinica*	0.15 (0.67)	0.99 (0.61)	0.14 (1.30)	0.14 (1.80)	-	2.70 (0.55)	4.12	II
EP05	*E. sinica*	-	0.32 (1.65)	-	-	-	-	0.32	III
EP06	*E. intermedia*	-	-	-	-	-	-	-	III
EP07	*E. przewalskii*	-	-	-	-	-	-	-	III
EP09	*E. przewalskii*	-	-	-	-	-	-	-	III
EP13	*E. lepidosperma*	-	-	-	-	-	-	-	III
EP14	*E. minuta*	-	-	-	-	-	-	-	III
EP15	*E. regeliana*	-	-	-	-	-	-	-	III
EP19	*E. przewalskii*	-	-	-	-	-	-	-	III
EP20	*E. intermedia*	-	-	-	-	-	-	-	III

## Data Availability

Original data can be obtained from corresponding author upon request.

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
