# Peer review of "A Modified 1H-NMR Quantification Method of Ephedrine Alkaloids in Ephedrae Herba Samples"

_ijms, 2023, doi:10.3390/ijms241411272_

Round 1
Reviewer 1 Report
In this multiauthored paper the authors give a detailed account on a simple 1H-NMR based quantification of ephedrine molecules in plant extracts. It also provides a detailed summary of previous results in the field.
In addition to canonic ephedrine structures shown in Fig.1 further cyclization products were identified in the diethyl ether extracts (lines 126-127 and Fig.2). It is not clear whether these are products not seen when using other solvents for extraction or else, may have formed by some chemical reaction during the extraction process? The authors should comment on this point.
What solvent was used for the detection of the NMR spectrum in Fig.6? Although it may be presumed from the context that it was deuterochloroform but this has to be stated explicitely in the caption to the figure.
In lines 232-233 reference is made to the „above-mentined equation 1”. Such formula is, however, apparently missing from the main text where it is being referred to.
Author Response
Response to Reviewer # 1
In this multiauthored paper the authors give a detailed account on a simple 1H-NMR based quantification of ephedrine molecules in plant extracts. It also provides a detailed summary of previous results in the field.
Response: Thank you for your comment. We wish this study would provide comprehensive knowledge of NMR quantification of ephedrine alkaloid analogs.
In addition to canonic ephedrine structures shown in Fig.1 further cyclization products were identified in the diethyl ether extracts (lines 126-127 and Fig.2). It is not clear whether these are products not seen when using other solvents for extraction or else, may have formed by some chemical reaction during the extraction process? The authors should comment on this point.
Response: Thank you for your comment. We had inserted one sentence “These products may be resulted from the chemical reactions among the ephedrine alkaloids and the impurities of diethyl ether” in lines 130-132. We hope this revision could make our sentences clearer.
What solvent was used for the detection of the NMR spectrum in Fig.6? Although it may be presumed from the context that it was deuterated chloroform but this has to be stated explicitly in the caption to the figure.
Response: Thank you for your comment. We added the deuterated solvent in the caption of Figure 6.
In lines 232-233 reference is made to the “above-mentioned equation 1”. Such formula is, however, apparently missing from the main text where it is being referred to.
Response: Thank you for your comment. We listed the equation 1 in the Experimental section. Therefore, we modify the sentence in line 231 as shown.
Author Response
Response to Reviewer # 2
The manuscript entitled "A Modified 1H-NMR Quantification Method of Ephedrine Alkaloids in Ephedrae Herba Samples" by Yue-Chiun Li et al. presents valuable results in the field of analytical chemistry. It is well-written and was carefully prepared. The reviewer recommends its publication and requests minor modifications according to the following remarks and questions.
Response: Thank you for your comment. We had modified the manuscript all according to your suggestions.
line 23. Ethyl ether is not the source of the problem, but its impurities. Please adjust the sentence.
lines 49 and 50. Too many figures after the decimal point.
line 83. First occurrence of the word Mahuang. Please explain here what this is.
lines 119 and 381. Please replace "ephedrine alkaloids" by "ephedrine alkaloid signals".
line 191. "To make sure the location of each alkaloid" could be replaced by "To ensure a correct NMR signal assignment for each alkaloid".
lines 208 and 211. Replace "resonated" by "resonating".
line 246. Please replace "experimental results" by "extracted alkaloids".
Response: Thank you for your comment. We had revised these minor errors according to your suggestion.
line 203. Why did you use CDCl3 as NMR solvent and not C6D6? CDCl3 could also create artefacts by reacting with sample alkaloids in the NMR sample tube, maybe slowly.
Response: Thank you for your question. Indeed, we also tried C6D6 as solvent (see the following figure), however, the NP and PE showed complete opposite locations compared with those in CDCl3. For the comparison with various literature published, we selected CDCl3 to record the NMR spectra of our samples. We totally agreed with your comment that CDCl3 could create artefacts, however, if we recorded the samples in C6D6, then all our data would be more difficult to compare with those already published.
line 235. Sorry, I do not understand "The average recovery of EP was about 95 %". To which recovery measurement does this make reference?
Response: We felt sorry for your misunderstanding. Here we measured the recovery of EP in triplicate, and the average data was about 95 %. All the experimental protocols were provided in Section 4.6.
line 309. Anthracene purity is 97 %. Was this fact taken into account in alkaloid content measurements?
Response: Yes, we had already counted this factor in our calculation model.
line 346. "100 scans were recorded in 0.187 Hz/point". Is this an FID resolution or a spectrum digital resolution?
Any zero-filling? Any digital filtering (apodization)? Number of complex FID data points? Number of complex spectrum data points? Please rephrase.
Response: Thank you for your comment. We’re sorry for listing these unclear parameters and the detail data have been modified at lines 348-350.
There is presently no requirement to deposit raw and transformed NMR spectra in public repositories but this is a good habit to adopt.
Spectra in pdf SI files are generally of no or little use.
Please take the time to read https://doi.org/10.1039/c7np00064b
The recommendations in https://doi.org/10.1186/s13321-021-00520-4 are also worth being followed and should apply to any Journal, not only J. Cheminform.
Response: Thank you for your comment. We will follow the direction of the present journal to keep or remove our supplementary data.